# An Exploratory Study of Nurses’ Feelings about COVID-19 after Experiencing SARS

**DOI:** 10.3390/ijerph20032256

**Published:** 2023-01-27

**Authors:** Hui-Ling Lee, Pei-Ju Chang, Li-Chiu Lin

**Affiliations:** 1Department of Nursing, University of Kang Ning, Taipei 114311, Taiwan; 2Chang Hua Hospital, MOHW, Changhua 513007, Taiwan; 3Nursing Department, Hong Kuang University, Taichung City 433304, Taiwan

**Keywords:** SARS, COVID-19, experience, epidemic, stress

## Abstract

The outbreak of severe acute respiratory syndrome (SARS) in 2003 in Taiwan impacted Taiwanese society. However, the first case of coronavirus disease 2019 (COVID-19) was reported in Wuhan and spread around the world. During these outbreaks, nursing staff experienced different levels of pressure. Studies have explored the stress and adjustment of nurses during these periods, but studies describing the feelings of nurses during both SARS and COVID-19 outbreaks are lacking. The aim of this study was to explore the experiences of nurses who had cared for both SARS and COVID-19 patients. A qualitative study combined with snowball sampling was applied. Semi-structured questions were used to interview 10 nurses who had experienced both SARS and COVID-19. Two themes and four sub-themes were analyzed, which were: facing the epidemic from the unknown to known; and the experiences from ignorance to proficiency. The sub-themes were: the feeling of frustration and concern; bottlenecks and pressures in my work; my mission and support; and positive energy and camaraderie. The results showed that the media acts as an important resource during disease outbreaks; therefore, government departments have to use their wisdom to make good use of the media. Secondly, understanding the general public’s response to the disease is also important for first-line nurses. Finally, on-the-job education and guidelines for first-line nurses are necessary, and support from the administration is also important.

## 1. Introduction

In 2002, an outbreak of severe acute respiratory syndrome (SARS) occurred suddenly. It caused a nosocomial infection in Taipei City Hospital, Heping Fuyou Branch. This incident had a significant impact on all medical staff in Taiwan at that time [1,2]. However, during that time, the media in Taiwan failed to fulfill its responsibility to assist the government in advocating policies [3] and tended to exaggerate and sensationalize the reports on SARS, that caused panic and anxiety among people [3].

Although the entire outbreak in Taiwan lasted only half a year, this period also resulted in the death of 11 medical staff [4]. Such a situation caused a wave of resignations of nursing staff. For the nursing staff who remained in the workplace, the situation caused a lot of emotional trauma and exhaustion [5]. As time goes by, the SARS epidemic is slowly forgotten in the hearts of the people.

However, in December 2019, a severe infectious pneumonia, coronavirus disease 2019 (COVID-19), suddenly broke out and spread all over the world. The global epidemic of COVID-19 is quite serious and has not yet eased. The media is constantly reporting on foreign epidemics, and due to the continuous mutation of the virus many countries have begun to implement city closure measures, with medical resources being almost exhausted, causing some nurses to be under intense pressure [6]. As of 24 December 2022, the number of confirmed cases worldwide was 651,960,325, and the global fatality rate was 1.02 % [7].

In Taiwan, because of the lessons learned from SARS in 2002, there was no large-scale infection in the early stage of the epidemic and most of the confirmed cases were immigrants, due to the gradual opening of the border for entry and the gradual relaxation of epidemic prevention policies. In May 2021 and May 2022, community cluster infections in Taiwan were also caught in the epidemic storm [8].

Compared with SARS, nurses facing COVID-19 have better equipment and related training, and especially nurses who have experienced SARS, have better coping methods for COVID-19. However, due to the long epidemic of COVID-19 and the death rate being higher than that of SARS, no country in the world has been spared. Social and family support, clear health policies, and relative benefits, are very important to first-line nursing staff.

## 2. Literature Review

During the outbreak of SARS 18 years ago, there were clusters of infections and Taipei City Hospital, Heping Fuyou Branch, decided to recall all personnel in the hospital for isolation. Because of the emergency closure of the hospital, relevant supporting measures, protective equipment, and the control of isolation moving lines were not fully established [1,2]. All medical staff, patients, and family members were exposed to high-risk cross-infection environments, causing them to have nowhere to seek help in the hospital, and they were deeply afraid of getting sick and dying [1,2]. The closure of the hospital, coupled with unclear policies, epidemic prevention measures adjusted at any time, and insufficient protective equipment, led to negative emotions and pressure on the nursing staff [9,10].

However, the working condition was not the only pressure to nurses, because of the media. As media had been called the “forth Estate”, it is powerful in affecting the public and has its social responsibilities to take [11]. Nevertheless, during the epidemic of SARS, the media mainly emphasized the “impact” dimension, with less concern in health education [11]. Lee et al. [12] (p. 353) also stated that in the period of SARS, the media coverage of the new of epidemic also reported statements which included: “nurses tried to escape from quarantined hospital”, ”hospital makes are in short supply make” or “disciplinary actions upon the hospitals failing to report SARS cases”, which caused nurses extreme stress and they experienced significant psychological conflict between their duties as a nurse, as well as their concern of their safety.

Because of the media, the first-line nurses were labeled, which indirectly affected the life of their families in the community, as well as their children’s learning at school [9,13]. Fortunately, the first SARS case was discovered in Taiwan on 14 March 2003. On 5 July 2003, the World Health Organization announced that Taiwan would be removed from the list of SARS-infected areas. The epidemic ended within four months [14].

The epidemic has been over for many years. Everyone’s work and life had returned to normal, but, at the end of 2019, COVID-19 began to spread. Due to the experiences learned from SARS, Taiwan was able to strictly control the epidemic in its early stages. However, hospitals and the public in Taiwan were also affected by COVID-19 [15]. COVID-19 caused clusters in communities in Taiwan, which transmitted to hospitals. The government and administration in hospitals decided to clear the hospitals instead of closing them. Once community infection spreads to the medical system, the biggest worry is the domino effect of hospital collapse, so the burdens of medical care have to be reduced [7,15]. However, COVID-19 has been going on for three years and has not stopped; it has caused a significant impact on society around the work needed [15].

Although in the face of COVID-19, with no end in sight, some studies have pointed out that nurses who have experienced SARS have made better adjustments in the face of COVID-19 [5]. Nurses who receive on-the-job education are able to reduce their fear [5], and, after adapting to the environment, negative emotions gradually reduce [16]. At the same time, a study by Fang et al. [1] also found that nurses who have experience in caring for patients with infectious diseases have lower psychological distress and better stress relief. However, a study by Huang et al. [13] found that when nurses who had experienced SARS had considerable knowledge of caring for the disease and sufficient equipment their stress and considerations shifted to matters related to themselves, such as worrying about whether they would infect their own family members.

Compared with SARS, nurses facing COVID-19 have better equipment and related training, and especially nurses who have experienced SARS, have better coping methods for COVID-19. However, due to the long epidemic of COVID-19 and the death rate being higher than that of SARS, no country in the world has been spared. Social and family support, as well as clear policies and relative benefits, are particularly important to first-line nursing staff.

## 3. Method

### 3.1. Research Design

In order to obtain the deep experience of the participants, this study adopted qualitative research to collect data in the form of in-depth interviews and semi-structured interview guidelines. The researchers used verbal and non-verbal communication modes to explore the feelings and responses of nurses who had cared for SARS patients prior to caring for COVID-19 patients.

### 3.2. Participants

A total of 10 participants were interviewed. The main research objects of this study were nurses who had experienced both SARS and COVID-19. Because the cases were not easy to obtain, the cases were collected by snowballing. We accepted cases until the data were saturated. The inclusion criteria were: (1) nursing staff who had directly cared for SARS and COVID-19 cases; and (2) nursing staff who could speak clearly and were willing to participate in the research. The basic information of the participants is shown in Table 1.

It can be seen from Table 1 that the participants were female nurses aged 42–55 (average age about 49 years old), with nursing experience of 21–35 years (average seniority about 24 years); nine were married and one was unmarried. During the SARS epidemic, they all worked in special units. Three participants were promoted to supervisor, but they still handled care-related business during the COVID-19 epidemic.

### 3.3. Data Collection

Snowball sampling combined with a semi-structured interview was applied in this study. The duration of data collection was between 2 February 2021 and 3 January 2022. However, in 2021, when the epidemic broke out in Taiwan, some participants were interviewed by communication software. The interview guideline was drawn up based on the purpose of the research and the literature review and included the following: (1) Can you talk about your experience when you faced the epidemic of SARS? (2) Regarding the COVID-19 epidemic, can you share your experience and feelings about it? (3) Can you talk about the differences between SARS and COVID-19? (4) Do you have any other things would like to share with me?

### 3.4. Data Analysis

In the data analysis, the narrations and extracted codes were discussed with other qualitative research experts in relation to the subject of study, and their indications were considered. An ethical consideration was obtaining permission from the ethics committee of the Guang Ten General Hospital Human Experiment Committee (IRB 10952), a written informed consent from the participants was obtained, and consideration was given to the participants’ right to cancel their attendance in the research.

### 3.5. Research Rigidity

To ensure rigor and credibility, this study followed the standards proposed by Lincoln and Guba [17]. Data credibility was established through the selection of appropriate data collection methods (guidelines for semi-structured interviews) and the manner in which the researchers conducting the interviews were familiar with the evidence of SARS and COVID-19. Reliability was ensured by describing the data analysis in detail and providing direct references to reveal the basis on which the analysis was performed. The researchers coded the interviews independently of each other. Consistency of analysis was determined through a meeting to discuss initial findings, where emerging norms and themes were discussed until a consensus was reached. This study was maintained throughout the coding process. To improve the portability of the study results, a background description, participant selection, data collection, and the analysis process are provided.

## 4. Results

This study mainly explored the feelings and responses of nurses who had experience in caring for SARS patients in the face of COVID-19. In total, 10 participants were interviewed by researchers. The basic information of the participants is shown in Table 1.

According to the data analysis, the researchers summarized the research results into two major themes and four sub-themes. The two main themes were: facing the epidemic from the unknown to the known; and the experience from ignorance to proficiency. There were four sub-themes: the feeling of frustration and concern; bottlenecks and pressures in my work; my mission and support; and positive energy and camaraderie. The details are shown in Figure 1.

### 4.1. Facing the Epidemic from Unknown to Known

Facing COVID-19 is a way to face new challenges by stepping on past experiences. The feeling was different compared to the experiences of SARS. Participants shared their feelings about facing COVID-19 after experiencing the outbreak of SARS. The researchers classified the relevant information into the feeling of frustration and concern and bottlenecks and pressures in my work, the details of which are presented in the following section.

#### 4.1.1. The Feeling of Frustration and Concern

In the face of the two epidemics, the participants felt worried and frustrated because they were not familiar with the disease; they also feared their mortality. Participants recalled the first time they faced the epidemic of SARS. The reason for their worry about SARS was fear:


*At the time of SARS...it was an unknown disease...we really did not want to join the team to take care of patients, because we were also afraid...and a group of colleagues was crying over there…scared because who knows what would happen...*
(Case H P2 L11)

SARS was an unknown disease to the participants. Case H also described their feelings when facing SARS, particularly fear. In addition, the information received through the media was mostly negative, which also brought pressure on Case I in terms of care:


*(SARS) I think the first thing I am not familiar with about this disease…it is an unknown condition, I don’t know how to control it…and then I heard a lot of negative news from the media, they (media) said we did not have enough equipment…they (media) a nurse pass away…from the news the pressure is relatively high…*
(Case I P3 L36)

As a result of interview, Case I indicated her fear was not only the unknown disease but also the information from media. As mentioned previously, some researchers had indicated that media had mislead the public and medical teams and caused panic. Fortunately, the period of SARS was only a couple of months and everything was under control. However, 18 years after experiencing SARS, COVID-19 appeared without warning. Participants indicated how they felt when they heard about the COVID-19 epidemic:


*…(SARS) Oh~ it’s finally over, because I feel like…oh my god! not again!…When (COVID-19) really came later, the whole thing was (gasp)...how could it be like this~ but just bite the bullet...but this time it feels like it’s nowhere in sight…*
(Case H P5 L23)

Case H decreased the intensity of their feelings when they faced COVID-19. The data of their interview showed that they had no choice and had less fear of the condition. Past experience allowed them to have a relatively low level of fear when facing the new epidemic. Case H pointed out with experience of SARS they had more protection for themselves when facing COVID-19. They said:


*(COVID-19) the whole epidemic is when it is more serious...anyway, I had experience (SARS) of it, and I am not afraid...I just told myself that this time the protection should be better, and the way of infection is different.*
(Case H P6 L36)

With prior experience, Case H shared her experiences of facing COVID-19. However, the occurrence of the nosocomial infection of COVID-19 and the infection of nursing staff, the spread of the media, and the reprimands of the public happened. In particular, the shadow of the hospital closure during the SARS period still affected Case D:


*…in fact, (SARS) Heping Fuyou Hospital was closed. …because of this (hospital closed), a young doctor died…Everyone was panic!...and (COVID -19) Butao (hospital)...happened again, people pointed the finger at the medical staff, and this is really hard to guard against~...It’s quite frustrating!*
(Case D P2 L33)

Whether it was the Heping Fuyou hospital closure during SARS, or the COVID-19 incident, the participants have had to bear a lot of pressure. They worried that the hospitals they serve would experience a similar situation. At the same time, the attitude and direction of public opinion would also change because of the conditions of the epidemics. The feelings of frustration can be seen in the content of the participants’ interviews. However, participants did not only feel frustrated by the condition of two epidemics and some misleading information; because of their duty, their families were also the participants’ biggest concern.

Participants mentioned that their family members tended to suggest they should resign their jobs. The participants also worried about infecting their family, especially those who were married with children. Case J mentioned that family members had asked them to leave their job due to concerns about the high risk of caregiving:


*…(SARS) at that time, my child was very young, and even the elder relative said, resign your job; it’s too dangerous…I’m also afraid that if I infect my family members,…the risk of infection is relatively high, the nurses left the job because their family wanted them to do it...*
(Case J P4 L27)

The unclear situation of SARS made the family worry about the safety of the participants. Conversely, the safety of the children at home was also a factor for participants to be concerned about not going home. In addition, there were nosocomial infections in both outbreaks, which made the participants worried that they might infect their family. The condition caused a lot of psychological pressure. Case F said:


*…(during SARS) We were sad because of the incident at the Heping Fuyou Hospital…we were afraid that we would infect our family, my kids were very young at that time...the pressure from the family was very high because you will say oh~...you see it is so serious...or…you might think of quit the job!*
(Case F P4 L10)

Because of SARS, the participants were concerned about cluster infection. However, when the COVID-19 epidemic broke out, participants learned that nursing staff and their families had been infected due to nosocomial infection. They felt more empathetic and feared that they would follow in the same footsteps:


*…This time (COVID-19) I saw the nurses (infected)...and their family were infected, and I felt really pitiful. I can understand that feeling...*
(Case B P12 L13)


*I did not go home because I did not want to cause my families infected…you know…during SARS, they (school) asked my child stay at home…they thought my kid was an infectious factor…early stage of COVID-19…you see from the news, almost the same…fortunately, this time the news from media was more acceptable…people changed their attitudes…*
(Case I P10 L10)

As a first-line nurse, participants had their duty. However, they had different feeling about the two outbreaks. Because of the unknown and the media’s misleading messages, participants felt panic during the time of SARS. In the face of COVID-19, due to different nursing experiences and disease attributes, the degree of fear was very personalized. However, family and children were their main concerns. Their work caused their families to be ostracized by the public or by children in schools. The results showed participants’ grievance and helplessness.

#### 4.1.2. Bottlenecks and Pressures in My Work

Whether it is SARS or COVID-19, it is a high-risk job for first-line nurses, who need to provide dedicated and centralized care for infected patients. The two epidemics in different periods caused different pressures for the participants because of the different care recipients; additionally, the pressure on patient care in the different epidemics was also different. For example, Case E indicates:


*…at that time (SARS),...working in the isolation room... work alone!...I was afraid of being infected...differently, (COVID-19)...we take care of patients with seriously condition, and the high infection rate...it is actually physical and mental stress…I’m tired...SARS was just a few months, but this time we really fought for too long!*
(Case E P5 L19)

Case E pointed out the differences of working with SARS and COVID-19. Working alone and a long period of fighting with epidemics caused them different pressures in their work. Apart from the duration of the epidemics, the participants also mentioned manpower and resources.

As previously mentioned, when the SARS outbreak occurred, there was no relevant information about the unknown disease. In particular, information on the disease and the policies were not very clear, so the participants were more likely to feel frustrated. The COVID-19 epidemic was based on the experience of SARS, and the command center set up by the government created all the information. The experience of SARS also made everything easier. Case A mentioned their experience of the two outbreaks. The first thing mentioned in Case A is the nursing manpower:


*…(SARS)…there was only one nurse on duty, there was no way to ask for help. Of course, inside the heart would become more anxious and panic...(COVID-19)…everyone will work together as a team…know how…know why…m! it’s much better.*
(Case A P20 L2)

Case A shared their experiences during the SARS and COVID-19 epidemics. They mentioned their working conditions in both epidemics. From their viewpoint, the manpower and cooperation with colleagues was important. Case D also expressed the importance of the team, which can support everyone through the situation together, and said:


*…(SARS)When you were there, you would be very lonely...everyone had to support each other, otherwise it is really hard, and everyone was panicked!*
(Case D P9 L3)

Teamwork was mentioned by many participants, and they thought it was an important factor to support the first-line nurses in continuing their work. In addition, it was indicated by some participants that relevant policies, procedures, and training, as well as on-the-job education, were particularly important and could alleviate the feeling of fear.


*…(SARS) at that time, I didn’t know how to use N95...to be honest, the equipment in our hospital was not enough, it was the first time, everyone has no experience!…no SOP at that time…and now (COVID-19)...education and training are quite frequent…It’s very important…I think it will really reduce the staff’s great pressure at work.*
(Case A P3 L30)

The experience of SARS made everyone aware of their shortcomings, so the planning of relevant training improved the basic capabilities of medical staff and enhanced their sense of security when caring for patients. In addition, Case C also indicated that on-the-job training was carried out regularly after the SARS incident to “prevent fools”.


*…After the SARS incident,...infection control will teach you! And they will give you a standard of picture to teach you how to put on and take of the protective suit, and take a test how to put on and take off every six months…we call it the “preventing fools”…*
(Case C P4 L5)

In addition to education and training, Case F also mentioned about the unknown information, lack of policies, and insufficient direction; they felt helpless because they were afraid of being infected. During the COVID-19 period, protective materials have improved significantly, and the protective equipment is sufficient.


*…it was chaotic at the beginning of SARS, there was no system, and when there was nothing to follow, everyone would be afraid...This time COVID-19,...our first-line nurses have first-hand information such as SOP, policies…I think it was not as frustrating as SARS...the protective measures are relatively complete.*
(Case F P2 L31)

The participants mentioned the importance of the policies and the training program, as well as sufficient manpower and equipment. With the accumulation of experience and the improvement of infection control measures, during this epidemic period, the hospital’s protective equipment was sufficient. With “know how” and “know why”, it is clear that information given to the first-line nurses means they have less pressure at work.

### 4.2. The Experience from Ignorance to Proficiency

When caring for patients during SARS, they felt ignorant, helpless, and afraid at the beginning, but due to their relevant experience their coping methods matured when facing COVID-19. From the data of the interviews, the researchers divided this stage into “my mission and support” and “positive energy and camaraderie”.

#### 4.2.1. My Mission and Support

From the participants’ point of view, sacrifice is the expectation and aura that nurses have always had. Many expectations have been placed on them. Since the outbreak of the SARS epidemic began in public hospitals, they could only accept instructions to take care of patients in the face of the epidemic. During COVID-19, the epidemic has been longer and more challenging compared to SARS, but, due to their relevant nursing experience, plus their passion for nursing, they are still willing to devote themselves to caring for patients:


*(SARS) the patients are very disturbed, and they will frequently ask, “Miss, am I severity?...Is it like the one reported on TV? Will I die? “...then you need to appease the patient’s feeling, and then you have to be strong enough…*
(Case A P3 L8)

During the interviews, Cases A and I also expressed that tourists carried the virus into Taiwan in the early stages of the spread of COVID-19. Faced with these patients, there was ambivalence, and they also figured out their problem of looking after people from overseas:


*…(COVID-19) I just want to say “Why should I care about you people?...foreigners come to Taiwan and carried the virus..., I feel...just...but I told myself that he is sick and is a patient, and he should be treated fairly, I don’t want to say that he is excluded. But do you know what’s my problem?...my language…*
(Case A P4 L9)


*…the foreigner...I don’t know…this time (COVID-19)…majority of the incidences were of COVID-19 were carried by foreigner, oh, yes, I did have to look after them but what caused me headache was language (laughing) I don’t think my English is good enough to communicate with them…*
(Case I P7 L19)

The participants described their commitment to the job. With their expectation and aura, they have to hide their fear and weakness from patients. They also have to hide their anger to face the tourist diagnosed with COVID-19. It is not easy and reflects the difficulty of those first-line nurses during epidemics. In addition, from the results of the interviews, the participants also presented the weakness of their language ability. This also caused the participants a feeling of pressure. However, in order to put themselves in a better condition to take care of their patients, they shared their experiences of how they adjusted themselves.


*Stress...I’m trying to face it positively...The first thing you should do is to be healthy! That is, the diet and nutrition must be sufficient, and then there must be enough sleep.*
(Case G P6 L36)

Case G said that eating well, sleeping well, and protecting themselves, were their coping methods to survive the epidemic. The participants also indicated that they monitored their health and maintained a positive attitude when facing the epidemic, as well as self-protection and washing their hands, which helped them feel at ease.


*…You have to learn how to protect yourself. If you feel that you have done a good job of protecting yourself, your mood will be more stable ~ feel at ease.*
(Case G P7 L4)


*I...I will try to avoid eating with my colleagues,...I think I have taken protective measures well, but I am not necessarily know whether my colleagues have done well or not…*
(Case F P9 L11)

Nurses know that their work risks are high, so they will protect themselves, including taking protective measures and washing their hands properly. In addition, in order to protect family members, the nursing staff will choose to clean themselves before going home or reduce contact with family members, as Case A said:


*(COVID-19)…We are afraid of infecting our family members, so before going home, we will take shower...*
(Case A P13 L9)

On the other hand, the support from family provides the nursing staff with great strength to help them through the epidemic period. The degree of support from family members determines whether the nursing staff are willing to continue to struggle in the workplace. They said:


*…I once told my husband that I am taking care of this kind of case, ...I am afraid of infecting him, but my husband told me “Don’t be afraid, if we are quarantined, we will be together”*
(Case A P14 L20)


*… my husband is very supportive of me...he believes that I will do well... My mother look after my children...During SARS, my family was very supportive, and then...when faced with such a situation (COVID-19) again, their attitude was still the same.*
(Case F P8 L2)

In the face of the two epidemics, the participants indicated that good management of health, including exercise, nutrition, and sleep, would help to prevent them being infected. Support from their family is the most important way to support nursing staff through the epidemic. Although their families are still worried for their safety, they also respect their profession and give them great support.

#### 4.2.2. Positive Energy and Camaraderie

As mentioned earlier, during the SARS period, due to the underdevelopment of communication software, many messages were unclear, and there was a feeling of isolation because of working alone. With the development of information nowadays, communication software and related information can be used to convey care or correct information, which can bring positivity.


*…ln the beginning (SARS) there were only mobile phones~ and then the phone...the feeling is lonely. ...nowadays technic is more developed~...it’s more...not feeling of isolated...*
(Case D P3 L25)


*(COVID-19) the power of the internet is quite strong... there will be social learning…there are some positive reports. ...everyone joins in the grand event...I think that it is much better than before.*
(Case I P12 L38)

As mentioned in Case I, the Internet is very powerful. The participants mentioned that they used the first-line group to convey care, and they also used the power of the Internet to spread a positive atmosphere. Compared with the SARS period, they have more positive energy to carry out their patient care and face the outbreak. When conveying messages using the Internet, the participants mentioned mutual help among colleagues.


*When I saw my colleagues, I cried!...They said, “why you are crying? We are here for you.”...you are not alone ….. I mean that the team can help each other~…*
(Case A P4 L25)


*…Colleagues...caring for each other, in fact, I also think it is very important to encourage each other...every time I go to the ward, I will remind each other to be carefully…*
(Case H P4 L6)


*…I think our dean is very kind. He leads us~ no criticize …everyone is working very hard in epidemic prevention, and we must give encouragement to these infected people. In fact, the hospital’s attitude towards us means that we should give care to these people and go through with them together.*
(Case G P19 L3)

Nurses on the first line often have to face many things. As mentioned by the participants, the dean’s approach of “no criticize” moved and warmed the participants. According to the results of the interviews, the participants shared their experiences and feelings when they faced the epidemics. It can be seen that support from the public and their families is very important to them. As well as the “no criticize” approach, these are key points that keep them going.

## 5. Discussion

From the results of the interviews, the participants indicated that, because of the unknown, people were afraid of SARS. For example, the participants said: *“Because it is an unknown disease, ...I am afraid because it is unknown”* and *“I feel panic! Not at all. I do know anything! I only know that it will kill people (SARS).*” This shows that when facing SARS people felt panic and fear because they did not know anything about the disease, only that there was a high mortality rate. As Maunder et al. [14] mentioned, the sources of stress for medical staff during the SARS period included uncertainty about the disease and increased mortality, which is consistent with the research results.

When faced with COVID-19, the feeling of frustration was reported by the participants. Due to previous nursing experience and the severity of the disease, the degree of and the reasons for the fear and stress were different. As the participants said: *“This COVID-19 is more terrifying than the previous one, because of the mutation”* and *“It may be because of SARS.”* However, they were less likely to feel so frustrated and overwhelmed. This shows that due to the participants’ experience of SARS, the degree of fear of COVID-19 is different; however, they still have the feeling of fear, worry, panic, and other negative feelings. The results of this study are consistent with the fact that many research studies have mentioned that there will be psychological distress during the epidemic [18,19].

In addition, family members are the most important factor affecting the participants. The participants mentioned that they are worried that they could be a source of infection to their family. For example, the participants said: *“If you have children, you will be more worried about whether you will infect the children...“* This reflects the research results that indicated the factor of worrying about infecting family members [5,9].

Moreover, during the SARS period, there were also people who wanted to leave their jobs because of their family members. For example, the participants said: *“The wave of resignation...many of them are because of parents’ concerns...because they think the risks of death are too high”* and *“At that time, my children were still very young, ...the elder relative said that it’s so dangerous, maybe I have to resign my job.”* During the COVID-19 period, there have been few resignations, which is different from the SARS period. Wei [20] mentioned that during the SARS period the nurses were under pressure and family members were reluctant to leave their jobs, which led to a shortage of manpower. Conversely, in the period of COVID-19, protecting and caring for first-line nurses is the most important mission for the administration [19].

According to the results of the interviews, the married participants had a high percentage of negative feeling about the epidemic compared to those who were unmarried, especially those with children. For example, the participants said: *“At that time (SARS)...I was younger... There are not too many things to be afraid of...Now that I have children~....”* Sun et al. [21] mentioned that caregivers with children at home will be particularly worried, but there are also studies pointing out that married people have a more positive attitude toward COVID-19 [21], which shows that this feeling is related to personal experience.

Moreover, regarding the participants’ experience during the SARS period, there were some factors that caused them to feel a kind of frustration, including working alone, insufficient protective materials, working with scant information, and unclear policies and lack of knowledge. Fortunately, those uncertain conditions have improved significantly during the COVID-19 period. For example, one participant said: *“(SARS) didn’t know how to wear the N95 at that time? ...there was no standard...education and training...like now (COVID-19)...this time it has been effectively improved.”* The participants’ reports reflect some researchers’ viewpoints. For example, Guo et al. [9] and Maunder et al. [14] pointed out that frequent policy changes and insufficient equipment are factors that cause first-line nurses physical and mental stress. In addition, Feng et al. [5] and Labrague et al. [22] stated that on-the-job education and accurate information about the disease can reduce first-line nurses’ fear and negative emotions.

A sense of mission has been held in Taiwanese nursing society since the 1900s until now. In this regard, all participants carry the sense of mission to care for patients from different parts of the world during epidemics. Many participants mentioned: *“I still have the responsibilities and missions…as a nurse, and I don’t want to say that they are excluded.”* and *“I continue to work in nursing work ~ mission drives me forward!”* The issues that nurses face were also indicated by Guo et al. in 2005. Their [9] study stated that, during the epidemic of SARS, uniforms are an honor for nurses but they also give them the responsibility and pressure to perform the job and care for patients.

Transcultural nursing practice is another challenge for participants. With the change in society, Taiwan has become a multi-culture country. COVID-19 has given the participants the pressure to face language problems. They maintained that: *“different languages are really the most stressful for us at present.”* In responding to participants’ pressure, Chuang et al. [23] suggested that in response to the advent of the era of globalization, Taiwan’s nursing profession should also step into internationalization. It has been suggested that the foreign language ability of nursing staff should be improved.

Moreover, the support from family has been most important for the first-line nurses during the epidemic. On the other hand, first-line nurses’ safety is the main concern of their families. Their families also respect the participants’ duty of profession and they provide great support. For example: *“my husband...he gave me a lot of psychological support”* and *“I think they support me a lot…from their hearts.”* Family support is the key factor for nurses in fighting epidemics, which has also been mentioned by Sun et al. [21]. In 2005, a study by Guo et al. investigated the work-related stress and coping behaviors during the SARS outbreak period among emergency nurses and found that the participants’ coping method was to feel and receive support from their family. The results of this study also reflect the viewpoints of Sun et al. [21] and Guo [9].

Apart from the support from family, mutual support among colleagues and positive encouragement from leaders and administrations create a positive atmosphere for participants: *“It’s very supportive the situation like this…colleagues are here with you…, and you feel very touched…”, “The emergency director...is very positive, ...very supportive of us.”* and *“The attitude by the hospital is that we want…our bosses be accompany with us. we are working together ….”* Pan et al. [10] mentioned the spirit of solidarity and the support from important others. Sun et al. [21] also mentioned that the hospital has a reward and welfare system to support and encourage nurses, and the encouragement of colleagues also makes nurses feel happy.

### 5.1. Limitations

The purpose of this study was to explore nurses’ feelings who had experienced both SARS and COVID-19. Therefore, the number of participants was limited. In addition, the experience of SARS was 18 years ago, so there may be some memory bias in the results.

Furthermore, the duration of the data collection was between 2 February 2021 and 3 January 2022. These participants had not experienced the outbreak of the epidemic in May 2022. The participants refused a second interview because a limit to speech was announced by the hospital administration. The investigation of nurses’ feelings during the May 2022 outbreak could be a further study for anyone who is interested.

### 5.2. Suggestions for the Future

Policy: a clear policy is very important for first line nurses. It also protects them to face to challenges of epidemics. It is seen that from SARS to COVID-19, the government agency in Taiwan had big improvements. However, in the early stage of COVID-19, the first line nurses and their families were still affected by the changeable conditions. As the result, the government agency has to think about the policies to support and protect the frontline soldier for the challenges in the unknown future.Equipment: sufficient epidemic prevention equipment will affect the willingness of first-line nurses to care for patients as well as the safety of first-line nurses. From SARS to COVID-19, the government agency and administration in hospitals might have to consider to prepare enough materials to prevent future challenges.Media: media are very important resources for publics especially during outbreaks. The media is a major source from which people perceive risk. As a result, the health authority should construct mechanisms or SOP to remind follow-up preventative of risk after these two evidences.Physical and psychological health management for first-line nurses: it is recommended that medical and psychological professional consultants, such as doctors or psychologists, are available to support the first-line medical team.On-the-job education: major institutions arrange infection control education and training regularly. It is recommended to internalize the education and training of wards and invite senior nursing staff with relevant nursing experience to serve as lecturers.Strengthen the foreign language ability and information retention of nursing staff: as society is more globalized, a multicultural nursing knowledge is necessary.

## 6. Conclusions

Two outbreaks have caused serious harm in the medical system and the general public’s health, especially with COVID-19 still unabated. Although the condition of an epidemic cannot be predicted, these two experiences can also be learned from, including the preparation of epidemic prevention materials and regular on-the-job education, especially relevant knowledge. Certainly, government decision-making capacity is also required.

In addition, the media is a useful resource, but it can also cause panic and stress among the population. The reaction of the public has a direct effect on nursing staff, and the reaction of the public is mostly based on the media. Therefore, proper media guidance is one of the best stress responses for nursing staff. The restriction and use of the media can test the wisdom of the competent authorities.

Finally, it is also necessary to consider supporting facilities related to the hospital, such as nosocomial infection prevention, sufficient manpower, and a rotation system, which can allow nursing staff to lower their levels of stress.

## Figures and Tables

**Figure 1 ijerph-20-02256-f001:**
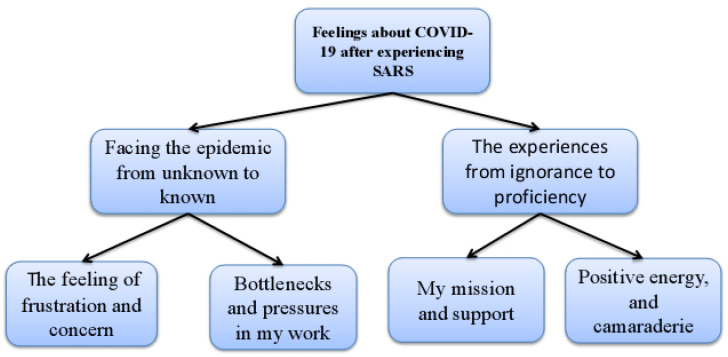
The themes and subthemes of study.

**Table 1 ijerph-20-02256-t001:** Demographic data.

No.	Age	Sex	Education	Marriage	Position		Total Seniority (Years)
					SARS	COVID-19	
A	43	F	College	Married	Nurse, Duty ward	Nurse, Isolation ward	22
B	55	F	University	Married	Nurse, ER	Nurse, OR	30
C	50	F	College	Married	Nurse, Duty ward	Nurse, Duty ward	28
D	52	F	Master	Married	Nurse, OR	Nurse, OR	30
E	42	F	University	Married	Nurse, ER	NP, ER	21
F	46	F	University	Married	NP, ER	NP, ER	25
G	50	F	University	Married	Nurse, Duty ward	HN, ND	29
H	44	F	University	Single	Nurse, Duty ward	Nurse, Duty ward	25
I	53	F	Master	Married	HN, Duty ward	SUV, ND	34
J	54	F	Master	Married	HN, Duty ward	HN, DR	35

## Data Availability

The information presented in this study is available on request from the corresponding author.

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
