# Peer review of "An Exploratory Study of Nurses’ Feelings about COVID-19 after Experiencing SARS"

_ijerph, 2023, doi:10.3390/ijerph20032256_

Round 1

Reviewer 1 Report

Although the topic of this paper is interesting, unfortunately it contains too many flaws and it will become publishable after a major re-write.  Details are contained in the attached report.

Author Response

  1. Although the topic of this paper is interesting, unfortunately it contains too many flaws and it will become publishable after a major re-write.  Details are contained in the attached report.

Authors response:

Thanks very much for the comments and kindly reminding. We are deeply appreciate your suggestions. The paper had been re-structured and seek language. In responding to other suggestions please see the following points, also please see the attachment of the changes. 

  1. The methodology described in a rather vague way; for instance, we have to wait till Table 1 to finally find out how many nurses took part in the experiment; Figure 1 has no title and is rather unclear;

Authors response:

In responding to reviewer’s concern, the figure 1 and the details of participants have been moved to page 5 line 151 to 158.

  1. The review of the literature is far from complete; it is more a narrative than a review of the literature as such

Authors response:

Thanks reviewer’s indication, the literature review has been revised. Please see page 3 (line 73-112); 4( line 73-133).

  1. There is a need for a serious linguistic revision; there are many mistakes, and some sentences are hardly comprehensible;

Authors response:

In responding to reviewer’s kindness reminding, the paper asked language help.

  1. The presentation of the results is much too long and includes too many quotes

Authors response:

In responding to reviewer’s concerns, the result had been re-organized again, please see page 6-14, line 192-492.

  1. The authors made no effort to demonstrate the link between the results and the recommendations

Authors response:

Thanks for reviewer’s reminding, the part of recommendation has been re-organized. Please see page 17-18, line 606-621.

  1. There are specific standards for the analysis of interview contents, and the authors don’t seem to be aware of them; the way in which they arrived at the two themes and four subthemes is unclear

Authors response:

The paper has been re-organized, please see page 6-14, line 192-492; and the themes and sub-themes of study is in page 7, line 207.  

  1. Other features are rather intriguing:

 Authors response:

Thanks for reviewer’s kindly reminding, in the section “Informed consent statement: page 19, Line 667”

  1. This study did not benefit from any funding? Where did the money come from to finance this research? The authors’ own pocket money?

Authors response:

Thanks for reviewer’s concern, however, this study, including the fee of IRB, all money were from authors. It had shown as page 19, Lin 662 mentioned.

Reviewer 2 Report

Dear authors,

The article presented is interesting for the scientific community. It covers a current topic that contributes to the advancement of knowledge on the state of the art.

However, authors should take into account the following observations to improve the quality of their work:

Firstly, it would be advisable to be more specific in the introduction section of the scientific manuscript.

A broader review of research related to nurses' feelings about COVID-19 is also recommended. In this regard, it is recommended that the Web of Science and SCOPUS databases be consulted.

On the other hand, a more in-depth theoretical analysis of the influence on healthcare workers of misinformation through the media in times of pandemic is suggested.

A more precise analysis of the conclusions drawn from the results obtained is also requested. It is necessary to better emphasize the originality and value of the research and then practical implications. 

Finally, the prospective or theoretical-practical implications of this research should be further developed and substantiated in order to analyse the usefulness of the study and its contribution to the scientific community.

Author Response

The article presented is interesting for the scientific community. It covers a current topic that contributes to the advancement of knowledge on the state of the art.

However, authors should take into account the following observations to improve the quality of their work:

 Authors response:

Thanks for reviewer’s comment and suggestion. In responding to all suggestions please see the following points and also check the attachments. 

  1. Firstly, it would be advisable to be more specific in the introduction section of the scientific manuscript.
    Authors response: the introduction has been re-organized, please see page2, line 37-70.

  1. A broader review of research related to nurses' feelings about COVID-19 is also recommended. In this regard, it is recommended that the Web of Science and SCOPUS databases be consulted.

Authors response: We deeply appreciate reviewer’s suggestion.  

  1. On the other hand, a more in-depth theoretical analysis of the influence on healthcare workers of misinformation through the media in times of pandemic is suggested.
    Authors response: In responds to reviewer’s suggestion, the concern of media has been added in different sections. Please check page 3 (line 84-101), page 7-8 (line 227-235)

A more precise analysis of the conclusions drawn from the results obtained is also requested. It is necessary to better emphasize the originality and value of the research and then practical implications. 
Authors response:

The paper has been re-organized, please see page 6-14, line 192-492.

  1. Finally, the prospective or theoretical-practical implications of this research should be further developed and substantiated in order to analyse the usefulness of the study and its contribution to the scientific community.

Authors response:

Regarding to reviewer’s concerns, this study indicated some suggestions including the policy, equipment, media control, physical and psychological health management for fist-line nurses, on-the-job education and strengthen the foreign language ability and information retention of nursing staff. Please see page 17-189, line 606-632.  

Reviewer 3 Report

The authors have employed a qualitative approach to understanding how the nurses felt and coped through 2 epidemics throughout their nursing careers. A number of valid points have been raised which I believe can be further expanded. 

Suggest the following:

1. The introduction was lengthy and a number of repetitions - can this be truncated to ensure brevity and clarity of the authors' thoughts

2. Extensive editing of the English language must be performed to allow the correct understanding of what Professor to be involved.

3. There were discussions on how the inaccurate portrayal of the disease by media may accelerate and result in negative emotions evoked in the public but other healthcare workers. This segment should be expanded with an enriched discussion on how the media should be moderated. 

4. There were also indications that demographic factors such as marital status and having children may affect how the nurses respond at work and deeper meaning for the job. It is hoped that this can be studied and further expanded.

Author Response

Reviewer #3:

Comments and Suggestions for Authors

The authors have employed a qualitative approach to understanding how the nurses felt and coped through 2 epidemics throughout their nursing careers. A number of valid points have been raised which I believe can be further expanded. 

Authors response:

Thanks very much for the comments and kindly reminding. We are deeply appreciating your suggestions. The paper had seek language help. In responding to other suggestions please see the following points.

Suggest the following:

  1. The introduction was lengthy and a number of repetitions - can this be truncated to ensure brevity and clarity of the authors' thoughts

Authors response:

Thanks for reviewer’s concern, the introduction has been re-structured. Please see page 2 line 38-71.

  1. Extensive editing of the English language must be performed to allow the correct understanding of what Professor to be involved.

Authors response:

In responding to reviewer’s concern, the paper has seek language help.

  1. There were discussions on how the inaccurate portrayal of the disease by media may accelerate and result in negative emotions evoked in the public but other healthcare workers. This segment should be expanded with an enriched discussion on how the media should be moderated.

Authors response:

 Authors response: In responds to reviewer’s suggestion, the concern of media has been added in different sections. Please check page 3 (line 84-101), page 7-8 (line 231-235), as well as page 18, line 618-621.

  1. There were also indications that demographic factors such as marital status and having children may affect how the nurses respond at work and deeper meaning for the job. It is hoped that this can be studied and further expanded.

Authors response: In responds to reviewer’s concern, please see page 8 ( line 265-268); 9 (line 269-306).

Round 2

Reviewer 3 Report

The authors have attempted to address all my concerns. However, English language can be further improved to enhance the clarity and improved understanding for the thoughts expressed by the authors.